# Degradation of Phenol Using Peroxymonosulfate Activated by a High Efficiency and Stable CoMgAl-LDH Catalyst

**DOI:** 10.3390/ma12060968

**Published:** 2019-03-23

**Authors:** Zhuwei Liao, Jingyi Zhu, Ali Jawad, Jiajing Muzi, Zhuqi Chen, Zhulei Chen

**Affiliations:** 1School of Environmental Science and Engineering, Huazhong University of Science and Technology, Wuhan 430074, China; liaozhuwei@outlook.com (Z.L.); zhujingyi1219@163.com (J.Z.); jawad_ali0@yahoo.com (A.J.); 2GAD Environmental Co., Ltd., Shenzhen 518067, China; muzijiajing@gmail.com; 3Key Laboratory of Materials Chemistry for Energy Conversion and Storage Ministry of Education, Hubei Key Laboratory of Materials of Materials and Service Failure, School of Chemistry and Chemical Engineering, Huazhong University of Science and Technology, Wuhan 430074, China; zqch@hust.edu.cn

**Keywords:** advanced oxidation processes, layered double hydrotalcite, hydrothermal method, reusability

## Abstract

In this study, we report on an active and stable CoMgAl layered double hydrotalcite (LDH) catalyst for phenol degradation by heterogeneous activation of peroxymonosulfate (PMS). The CoMgAl-LDH catalyst was synthesized by hydrothermal method. The PMS/CoMgAl-LDH system overcomes the drawbacks of traditional Fenton processes. Various effects, e.g., scavengers, chloride ion, catalyst dosage, PMS concentration, temperature, and pH, were also inspected to evaluate the system. The results indicated that the PMS/CoMgAl-LDH system had extremely high efficiency for phenol degradation; 0.1 mM phenol could be completely degraded by 0.3 g/L catalyst and 3 mM PMS within 60 min at 30 °C. The CoMgAl-LDH catalyst appeared to possess outstanding reusability and stability. After four rounds of recycling, nearly 100% of the phenol was removed within 80 min by the PMS/CoMgAl-LDH system, with only 0.05 mg/L Co^2+^ leaching. A sulfate radical was the main oxidation species in the PMS/Co-LDH system. The degradation rate of phenol was influenced by temperature, and the activation energy was 65.19 kJ/mol. These advantages proved the PMS/CoMgAl-LDH system is an effective strategy for the treatment of organic contaminants.

## 1. Introduction

With the reduction of freshwater resources, researchers have been prompted to recover water from industrial or domestic wastewater [1]. Among various wastewater treatment technologies, advanced oxidation processes (AOPs) are an effective and attractive technology for completely degrading organic compounds by reactive oxygen species generation, such as hydroxyl radicals (•OH) [2,3,4]. Various AOPs technologies, like ozone, UV, electro-Fenton, and sono-Fenton, have been developed [5], but extra energy, strict pH limitation, metal ion release, and sludge by-products have made AOPs economically costly and environmentally unfriendly [6].

Recently, research of AOPs technology into sulfate radicals (SO_4_^•−^) generated by peroxymonosulfate (PMS) has increased [7,8]. This is due to sulfate radicals having higher standard redox potential (2.5–3.1 V) than hydroxyl radicals (2.2–3.1 V) as well as possessing a longer half-life (30–40 μs) [9]. Moreover, PMS is more stable than H_2_O_2_, thus easier to store and transport [10]. PMS alone shows very low oxidation potential toward organic contaminants in solution. Energies like UV, microwave, and ultrasound are used for PMS activation [11], but extra energies limit the application of these technologies. Catalytic treatment by transition metals appeared to be an ideal approach for activation of PMS; homogeneous Co^2+^ was recognized as the most efficient reagent for PMS activation, but toxic metal ion residues were an important shortcoming of this technology [12]. Cobalt oxides and cobalt supported heterogeneous catalysts appeared to perform as well as Co^2+^; however, the Co^2+^ leaching of these heterogeneous catalysts was unsolved [13,14].

In this aspect, heterogeneous cobalt catalysts supported on layered double hydroxides (LDH) become a competitive alternative. The LDH structure consists of positively charged stacked brucite-like octahedral hydroxide layers with tunable M(II)/M(III) [15]. Thus, a transition metal can be introduced within the layered structure, which offers remarkable dispersion and more strong linkages among active sites [16]. This highly dispersed mixture of metal oxides makes LDH an ideal material of support for heterogeneous catalysts [17]. Besides, the basic character, memory effect, and good thermal stability are the additional distinguishing features associated with LDH based catalysts. Co-LDH catalysts appeared to possess outstanding efficiency and stability in bicarbonate activated H_2_O_2_ system, in contrast to the acidic conditions of traditional Fe^2+^-Fenton, H_2_O_2_/Co-LDH systems, indicating the performance could maintain in a wide pH range of 5–12 by diverse free radicals [18]. The co-precipitation method is the most commonly used synthetic technique for LDHs, which use M^2+^ (or M^2+^ mixtures) and M^3+^ (or M^3+^ mixtures) aqueous solutions as precursors [19]. However, the co-precipitation method is often accompanied by some poor crystallinity areas [20]. The hydrothermal method uses a stainless steel autoclave under high pressure allowing materials have better crystallinity [21]. Urea is an attractive agent for precipitation, which can precipitate several metal ions [22]. 

In this work, a CoMgAl-LDH catalyst was synthesized by the hydrothermal method, and the physicochemical properties of the catalyst were characterized. The PMS/CoMgAl-LDH system displays remarkable performance and excellent stability in phenol degradation. The effects of radical scavengers, chloride ion, catalysts dosage, PMS concentration, temperature, and pH during reactions were evaluated. Moreover, the reusability of the CoMgAl-LDH catalysts was also studied.

## 2. Materials and Methods

### 2.1. Chemicals

Phenol, PMS, tert-butyl alcohol (TBA), ethanol (EtOH), and urea were obtained from the No.3 Reagent Factory (Wuhan, China). Co(NO_3_)·6H_2_O, Mg(NO_3_)_2_·6H_2_O, Al(NO_3_)_3_·9H_2_O, NaCl, and other chemicals were purchased from Sinopharm Chemical Reagent Co., Ltd. (Wuhan, China). All chemicals were used as received without further purification.

### 2.2. Synthesis of Catalyst 

The CoMgAl-LDH catalyst was synthesized by the hydrothermal method. Co(NO_3_)·6H_2_O (0.1 M), Mg(NO_3_)_2_·6H_2_O (1.5 M), Al(NO_3_)_3_·9H_2_O (0.8 M), and urea were dissolved in deionized water at room temperature and transferred into a 500 mL stainless steel high-pressure reactor, sealed, and maintained at 140 °C for 24 h. The precipitates were filtered and washed four times and dried at 90 °C for 12 h, and were denoted as CoMgAl-LDH. CoAl and MgAl catalysts were prepared by the same method for comparison.

### 2.3. Experimental Procedure

#### 2.3.1. Batch Reaction

Experiments were conducted in a 30 mL glass beaker with phenol (20 mL, 0.1 mM). Three millimolar PMS and 0.3 g/L CoMgAl-LDH catalyst were added to the beaker. The beaker was stirred by a magnetic stirrer (500 rpm) at 30 °C. Two milliliters of the solution were withdrawn at 10, 20, 40, 60, and 80 min. The samples were filtered by a 0.2 μm filter and quenched with EtOH for analysis. The initial pH was adjusted with 1 M NaOH or 1 M H_2_SO_4_. Radical scavengers or chloride ions of appropriate amounts were added before the addition of PMS.

#### 2.3.2. Analysis 

The degradation of phenol was analyzed by high-performance liquid chromatography (HPLC FL–2200, Agilent, Santa Clara, CA, USA) equipped with a column (4.6 × 250 mm, 5 μm) and a UV wavelength of 280 nm. The volume ratio of methanol and water (60/40, *v*/*v*) was used as the mobile phase at a constant flow rate (1 mL/min). The concentration of leaching Co^2+^ was measured by microwave plasma atomic emission spectroscopy (4100 MP–AES, Agilent, Santa Clara, CA, USA).

### 2.4. Characterization Techniques

The structure of the catalyst was measured by X-ray diffraction (XRD, X’Pert PRO, Malvern Panalytical, Almelo, Holland) with a diffractometer of Cu Kα radiation (λ = 1.54 Å). The morphology of the catalyst was observed from pictures taken on a Quanta 200 scanning electronic microscope. Surface area measurement of the catalyst was carried out on a Micromeritics ASAP 2020 analyzer (Norcross, GA, USA). Before low-temperature physisorption of N_2_, the catalyst was dried and degassed at 120 °C under vacuum for 1 day. Metal oxidation states of the catalyst were measured with X-ray photoelectron spectroscopy (XPS, ESCALAB Mark II, Thermo Fisher Scientific, MA, USA) using standard and monochrochromatic sources (Al Kα), with the binding energy calibrated by the carbon signal at 285 eV. XPS spectra were analyzed using XPSPeak (4.1, Raymund Kwok, Hong Kong, China).

## 3. Results and Discussion

### 3.1. Characterizations

Figure 1A shows the XRD patterns of the CoMgAl-LDH catalyst. Rhombohedral symmetry with well-formed layered structures were demonstrated by the (001), (003), and (006) planes at 2θ ≈ 11°, 23°, 35° (JCPDS-022-0700). Peaks at 2θ ≈ 39°, 47°, 61°, and 62° (JCPDS-089-0460) indicated (015), (018), (110), and (113) crystal planes of LDH compounds, respectively, which were similar to those reported by Reference [23]. Typical hexagonal LDH symmetry can be seen in the SEM image (Figure 1B) of CoMgAl-LDH particles. The specific surface area of the CoMgAl-LDH catalyst was 92.51 m^2^/g. The oxidation states of the transition metal were related to catalytic activity. XPS was conducted for the oxidation states of Co (Figure 1C). The Co 2P_3/2_ peak was located at 780.9 eV, with a satellite peak at 785.6 eV. The spin-orbital splitting between the Co 2p_3/2_ and Co 2p_1/2_ peaks is 16.0 eV. According to the Co 2P_3/2_ binding energy and separation between components of the Co 2p doublet, the cobalt-containing material may be Co(OH)_2_ [24]. The results indicated the oxidation states of cobalt as Co^2+^ and excluded the existence of Co^3+^.

### 3.2. Degradation of Phenol

The phenol degradation performance of the PMS/CoMgAl-LDH system is shown in Figure 2. CoMgAl-LDH had similar catalytic activity to its equivalent of Co^2+^ solution, which totally removed phenol in 80 min. It was reported 0.3 mM Co^2+^ can strongly activate PMS (0.6 mM) to generate sulfate radicals that degrade 90% of organic pollutants (0.3mM) in water [25]. We compared the Co^2+^ concentration of the solution after the reaction and found there was virtually no Co^2+^ (0.01 mg/L) in the heterogeneous PMS/CoMgAL-LDH system, but there were 0.41 mg/L Co^2+^ in the homogeneous PMS/Co^2+^ system. The results indicate that both the homogeneous and heterogeneous reaction of Co could efficiently activate PMS, but the heterogeneous PMS/CoMgAL-LDH system could effectively avoid cobalt ion leaching. 

For other catalysts such as Co_3_O_4_, MgO, Al_2_O_3_, CoAl, and MgAl, the removal of phenol was 58%, 1%, 5%, 75%, and 15%, respectively. The degradation performance of these catalysts was much lower than CoMgAl-LDH catalyst, and the relatively good catalytic activity of Co_3_O_4_ and CoAl due to Co^2+^ leaching during the reaction. This result clearly demonstrated the excellent phenol removal performance of the PMS/CoMgAl-LDH system was due to the CoMgAl-LDH catalyst.

### 3.3. Influence of Radical Scavengers and Chloride Ions

In general, sulfate radicals were considered to be the main oxidant species in the PMS-based system. However, in alkaline conditions, sulfate radicals could react with OH^−^ to generate hydroxyl radicals [26,27]. Quenching experiments were performed to identify the main free radicals in the PMS/CoMgAl-LDH system. EtOH and TBA were used as selective scavengers to differentiate between SO_4_^•−^ and •OH radicals, respectively [9]. According to Figure 3A, by adding TBA, the phenol degradation showed no obvious changes in the PMS/CoMgAl-LDH system, indicating no •OH radicals were generated during the reaction. When adding 0.1 M EtOH, the phenol degradation decreased significantly to 40%. The inhibitory effect was further enhanced by increasing the amount of ethanol. By adding 3 M EtOH, the phenol removal efficiency was nearly 0%, which demonstrated sulfate radicals were the only oxidant species. XPS analysis of CoMgAl-LDH revealed the Co^2+^ states. In the PMS/CoMgAl-LDH system, the activation of PMS via the catalyst is proposed as the following steps (Equation (1)):Co^2+^ + HSO_5_^−^ → Co^3+^ + SO_4_^•−^ + OH^−^(1)

Researchers have not yet determined the influence of chloride ions on the PMS based oxidation reaction [28]. In general, Cl^−^ could be oxidized by SO_4_^•−^ to form a chlorine radical (Cl^•^), which would combine with another chloride becoming Cl_2_^•−^. The standard redox potential of Cl^•^ (2.09 V) was lower than SO_4_^•−^ (2.5–3.1 V), which reduced reactivity [29]. In the PMS/CoMgAl-LDH system, the phenol degradation was slightly accelerated by the addition of Cl^−^ (Figure 3B). For instance, nearly 100% of phenol was removed at 40 min when the amount of Cl^−^ was above 3 mM. Moreover, when adding 3 mM and 15 mM Cl^−^, the k_obs_ of phenol was increased 29% and 31% compared to no Cl^−^ in the system, respectively (Figure 3C). However, by further increasing the concentration of Cl^−^, the rate of k_obs_ growth began to reduce, and when the concentration of Cl^−^ was 30 mM, the k_obs_ of phenol was only increased 10% compared to no Cl^−^ in the system. Moreover, when the concentration of Cl^−^ was 90 mM, the k_obs_ of phenol was equal to the initial state. These results indicated the reaction was accelerated when there was a small amount of chloride ions in the PMS/CoMgAl-LDH system, and the system was not affected by the presence of excessive chloride ions.

### 3.4. Effects of Other Parameters

#### 3.4.1. Effect of CoMgAl-LDH Dosage 

The influence of different catalyst dosages (0, 0.2, 0.3, and 0.4 g/L) for phenol degradation was evaluated in Figure 4. The amount of PMS (3 mM) and phenol (0.1 mM) was fixed. The results indicated that only 10% of phenol was removed by PMS without any catalyst. The performance of phenol degradation was suddenly increased to 88% in the presence of 0.2 g/L catalyst. The removal of phenol reached 100% at 0.3 g/L of catalyst. The results indicated the active sites for PMS activation increased as the catalyst dosage increased. The presence of catalyst was favorable for PMS to generate sulfate radicals and increased the performance of phenol degradation [30]. However, by further increasing the amount of the catalyst, phenol degradation efficiencies appeared to be slightly enhanced.

#### 3.4.2. Effect of PMS Concentration 

The influence of different concentrations of PMS (0, 2, 3, and 4 mM) for phenol degradation was evaluated in Figure 5. The amount of CoMgAl-LDH (0.3 g/L) and phenol (0.1 mM) was fixed. The results indicated that CoMgAl-LDH catalyst alone had limited reactivity, about 5%. However, when 2 mM of PMS was added, the phenol degradation abruptly increased to 65% in 80 min. By further increasing PMS concentration, phenol degradation was accelerated; 100% phenol removal appeared at 60 min after increasing the amount of PMS to 3 mM and 4 mM. More free radicals could be generated at higher PMS concentrations, which may have contributed to the high phenol degradation rate phenomenon.

#### 3.4.3. Effect of Temperature

The influence of different temperatures (20, 30, and 40 °C) for phenol degradation is shown in Figure 6A. The amount of CoMgAl-LDH (0.3 g/L), PMS (3 mM), and phenol (0.1 mM) was fixed. The phenol degradation improved from 83% to 100% when the temperature was increased from 20 °C to 30 °C. However, 98% of phenol was removed in 40 min with no further increases when the temperature was 40 °C. Adverse effects owing to the competitive consumption of radicals with high temperature and the diffusion limitation phenomenon were reported [31].

The first order kinetics reaction rate constants were calculated in Table 1. The correlation between the constants and temperature was fitted with an Arrhenius relationship (Figure 6B). The activation energy of the reaction was 65.19 kJ/mol, which was higher than the diffusion-controlled reaction (about 10 to 13 kJ/mol) [32], indicating that the reaction rate was attributed to chemical reactions on the surface.

#### 3.4.4. Influence of pH

The efficiency of PMS based AOPs technologies is generally dependent on the initial pH of the solution [33]. The influence of initial pH was investigated over a wide range (pH 3.0–11.0, not buffered). As shown in Figure 7A, phenol degradation was accelerated at pH 3, 5, and 11 when compared with neutral conditions (pH = 6), but the performance of the PMS/CoMgAl-LDH system decreased at pH 9. The pK_a_ of HSO_5_^−^ was 9.4 [34], so it can be assumed that HSO_5_^−^ was the only PMS species in solution under neutral and acidic condition (pH 3, 5, and 6), while a small amount of SO_5_^2−^ was in the solution under alkaline conditions (pH 9 and 11). The pH_pzc_ of CoMgAl-LDH was valued at about 7.7. When the pH of the solution was higher than the pH_pzc_ of the catalyst, the surface of the catalyst was negatively charged. Therefore, the interaction between the catalyst and SO_5_^2−^ decreased the degradation efficiency when the initial pH was 9. The pk_a_ of phenol was 10. When the initial pH was 11, phenol existed in its deprotonated form with a positive charge, which means the charge between the catalyst and phenol was mutually exclusive. In theory, the reactivity should have been relatively weakened at pH 11, but the performance of the PMS/CoMgAl-LDH system was enhanced, which indicated that there are other factors activating PMS beyond the CoMgAl-LDH catalyst. We speculated that PMS was activated by alkaline conditions, and this theory was confirmed in Figure 7B; when the initial pH was 11, phenol degradation achieved 100% in 80 min with PMS alone. In acidic conditions (pH 3 and 5), the leaching of Co^2+^ after reaction was 0.39 and 0.13 mg/L, respectively, and in neutral or alkaline condition, nearly no Co^2+^ (<0.01 mg/L) could be found in the solution. When the initial pH was 3, only using PMS for phenol degradation was less than 10% (Figure 7B); the increase in phenol degradation may have been due to the presence of the PMS/Co^2+^ system.

### 3.5. Reusability of CoMgAl-LDH Catalyst

For the reusability evaluation of the CoMgAl-LDH catalyst, after each reaction, the catalyst was filtered and washed four times and dried at 90 °C for 12 h. As can be seen from Figure 8A, after four times of reuse, the CoMgAl-LDH retained excellent catalytic performance, which activated PMS to remove nearly 100% of the phenol in 80 min. Although the catalyst can maintain high activity during repeated use, the reaction rate gradually decreased as the number of cycles increased (Figure 8C). For instance, the k_obs_ for phenol degradation declined by 12.5% in the second round, and as recycling continued, k_obs_ gradually decreased by 57% in the fourth round. However, after four cycles, it was found that only 0.05 mg/L of Co^2+^ remained in the solution (Table 2). This demonstrated that the CoMgAl-LDH catalyst not only had exceptional reusability, but also appeared outstandingly stable.

## 4. Conclusions

In summary, we reported on a highly efficient, stable, and reusable PMS/CoMgAl-LDH system. CoMgAl-LDH catalyst synthesized by the hydrothermal method presented high catalytic activity in the activation of PMS for phenol degradation. Furthermore, virtually no Co^2+^ remained in the solution after reaction. Further investigations indicated that phenol degradation was affected by various parameters, such as catalyst dosage, PMS concentration, and temperature. The phenol degradation followed well with first order kinetics, and the activation energy was 65.19 kJ/mol. The PMS/CoMgAl-LDH system maintained highly performance in the initial pH range of 3 to 9, and the system was not affected by the presence of excessive chloride ions. The CoMgAl-LDH catalyst showed high reusability; after being recycled four times, the system could nearly achieve 100% phenol degradation with rare cobalt ion leaching.

## Figures and Tables

**Figure 1 materials-12-00968-f001:**
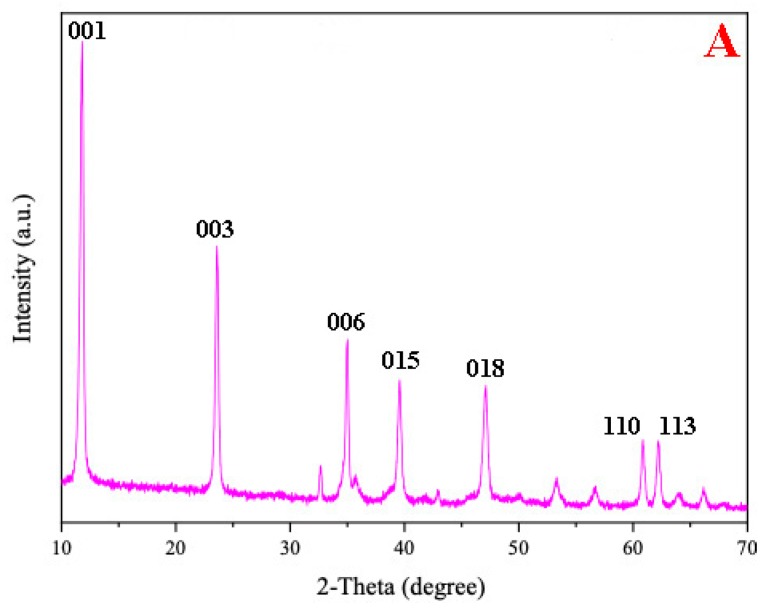
(**A**) X-ray diffraction (XRD) patterns of CoMgAl-LDH, (**B**) scanning electron microscope (SEM) images of CoMgAl-LDH, and (**C**) X-ray photoelectron spectroscopy (XPS) analysis of CoMgAl-LDH.

**Figure 2 materials-12-00968-f002:**
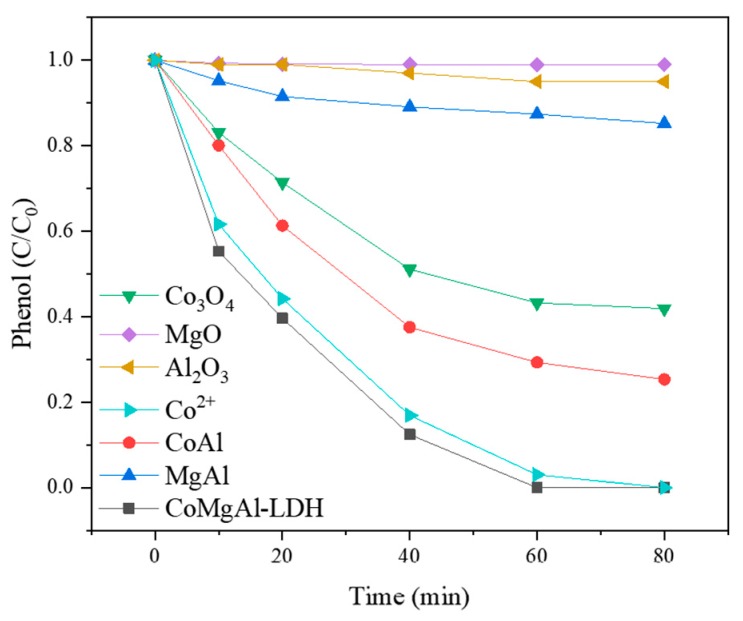
Degradation of phenol with CoMgAl-LDH, family oxides and compounds, and cobalt ions.

**Figure 3 materials-12-00968-f003:**
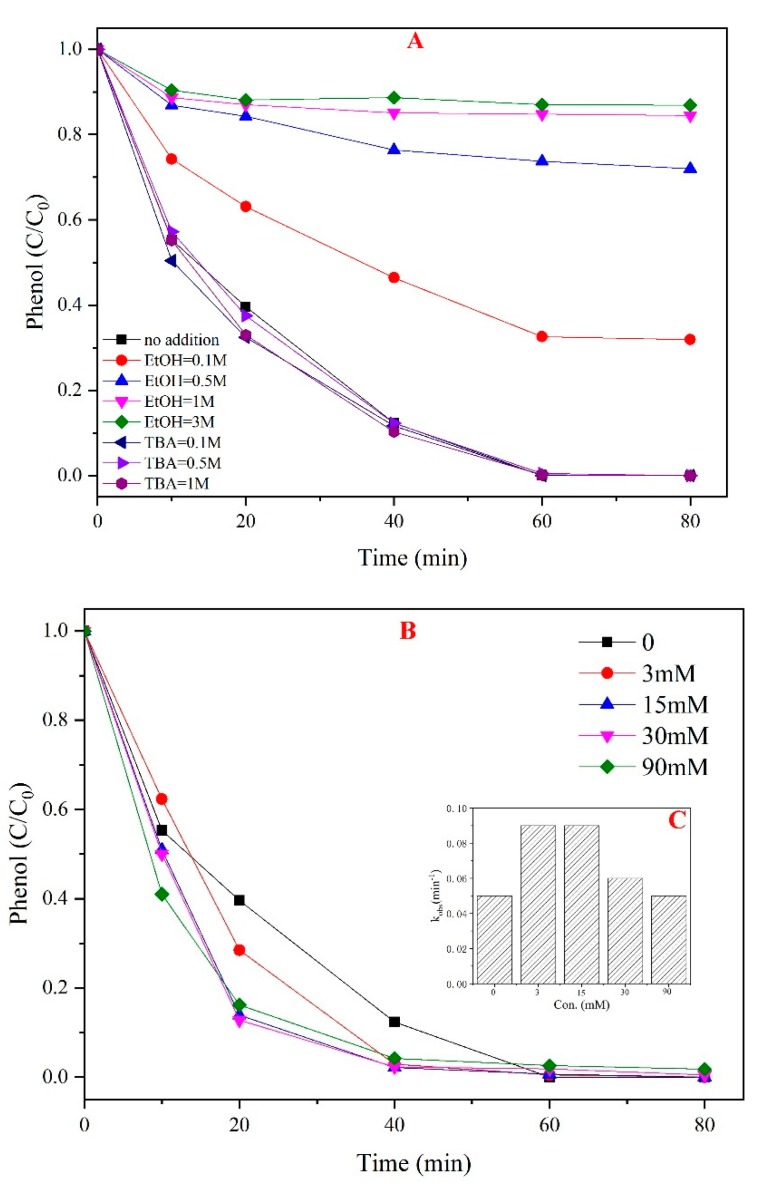
Effect of scavengers and chloride ions. (**A**) Effect of EtOH and TBA, (**B**) different concentration of Cl^−^, (**C**) k_obs_ of PMS/CoMgAl-LDH system.

**Figure 4 materials-12-00968-f004:**
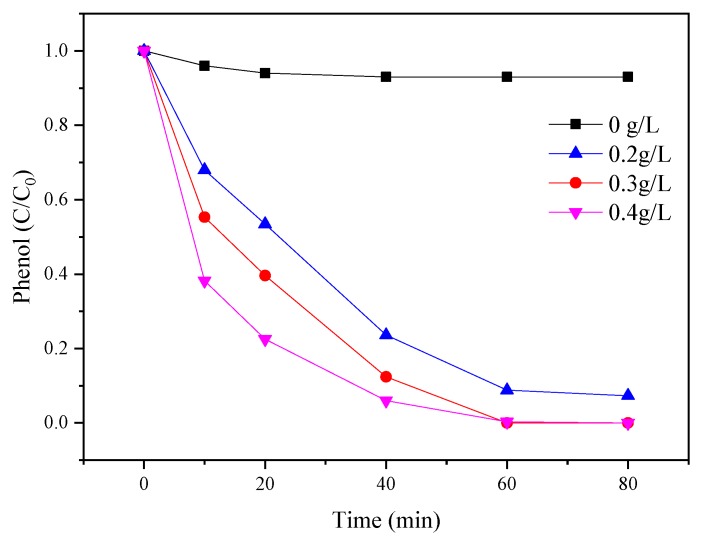
Effects of catalyst dosage on the PMS/CoMgAl-LDH system during phenol degradation.

**Figure 5 materials-12-00968-f005:**
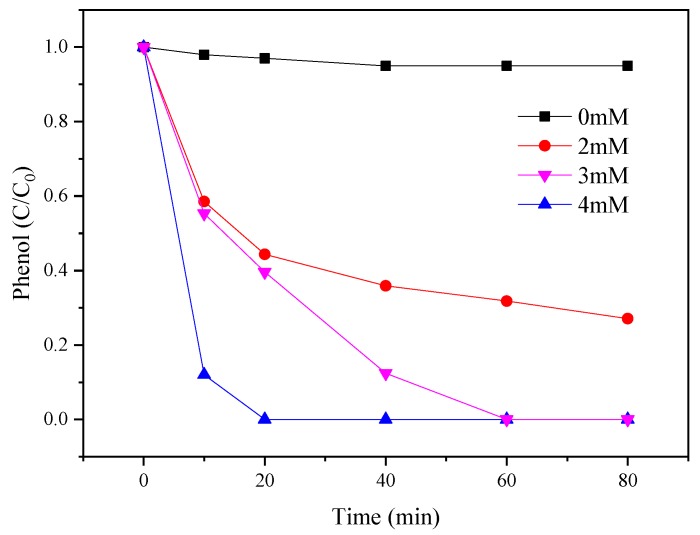
Effects of PMS concentration on the PMS/CoMgAl-LDH system during phenol degradation.

**Figure 6 materials-12-00968-f006:**
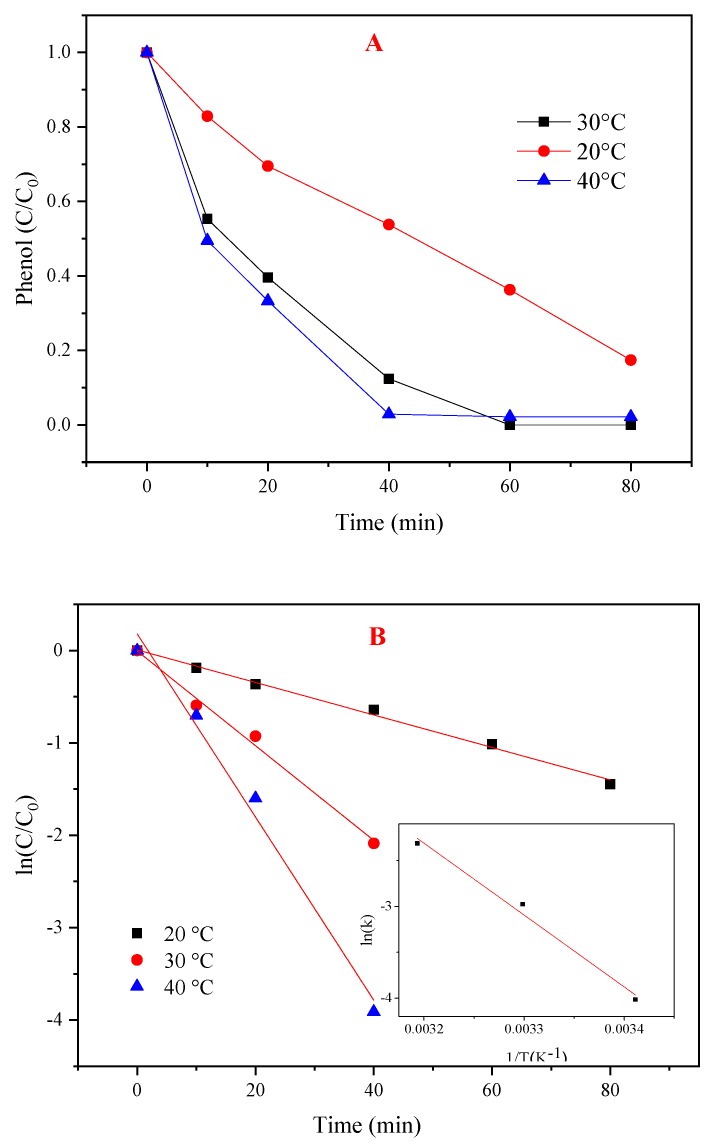
Effects of temperature on the PMS/CoMgAl-LDH system during phenol degradation. (**A**) Different temperature, (**B**) ln(C/C_0_) versus reaction time.

**Figure 7 materials-12-00968-f007:**
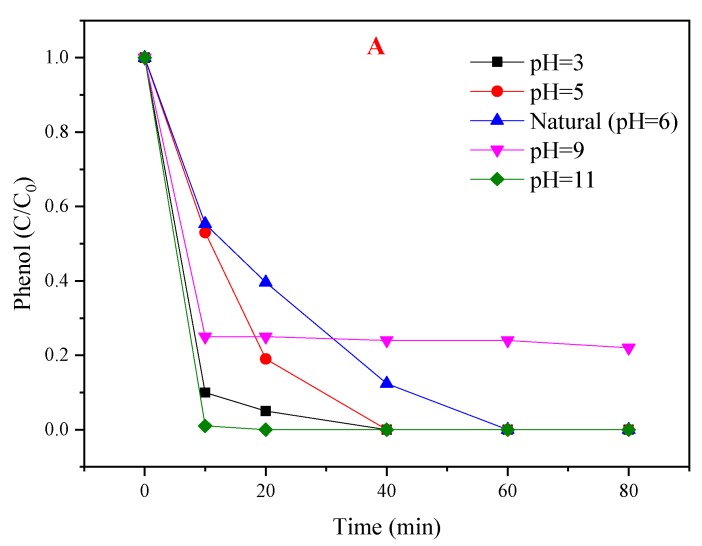
(**A**) Influence of different initial pHs of the PMS/CoMgAl-LDH system during the degradation of phenol; (**B**) comparison of absence and presence of CoMgAl-LDH with PMS removal of phenol.

**Figure 8 materials-12-00968-f008:**
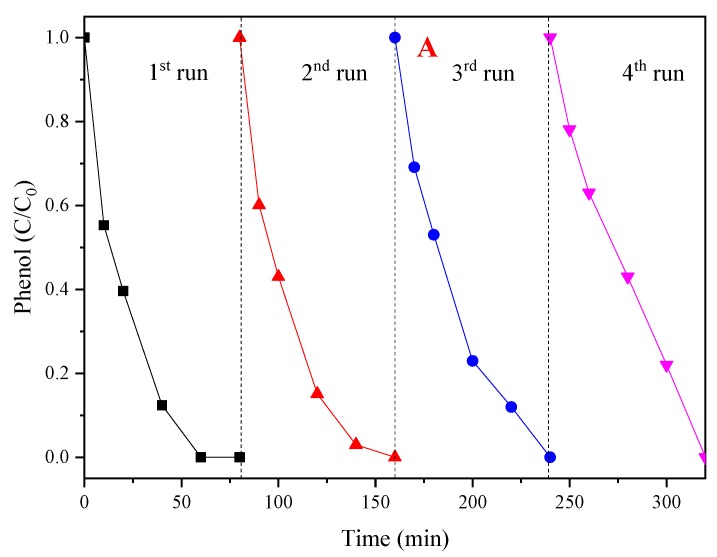
(**A**) The reusability of the CoMgAl-LDH catalyst; (**B**) ln(C/C_0_) versus reaction time based on the effect of recycle reaction; (**C**) k_obs_ for phenol degradation at various cycles.

**Table 1 materials-12-00968-t001:** The first order kinetic parameters for phenol degradation by the PMS/CoMgAl-LDH system at various temperatures.

t (°C)	k_obs_ (min^−1^)	R^2^
20	0.018	0.99
30	0.051	0.99
40	0.099	0.98

**Table 2 materials-12-00968-t002:** Co^2+^ leaching of CoMgAl-LDH after each cycle.

Cycles	Co^2+^ (mg/L)
1	0.01
2	0.01
3	0.03
4	0.05

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
