# Peer review of "Degradation of Phenol Using Peroxymonosulfate Activated by a High Efficiency and Stable CoMgAl-LDH Catalyst"

_materials, 2019, doi:10.3390/ma12060968_

Round 1

Reviewer 1 Report

I recommend that authors need to read their manuscript carefully before submission. There are so many errors, hence actually I cannot evaluate the manuscript.

For examples

Line 12-13 at page 1.  

Fenton process, and .?

Line 136 at page 5 is not relevant to the content of Table 1.

The condition of the catalytic experiment is improperly written.

So many serious flaws in the manuscript.

Author Response

Response to Reviewer 1 Comments

Dear Reviewer:

Thank you for your comments concerning our manuscript entitled “CoMgAl-LDH Synthesized by Hydrothermal Method Catalyzed Peroxymonosulfate with High Stability and Reusability for Degradation of Phenol” (ID: 435643). Those comments are all valuable and very helpful for revising and improving our manuscript, as well as the important guiding significance to our researches. We have studied comments carefully and have made correction which we hope to meet with approval. The main correction in the paper and responds to reviewer’s comments are as flowing:

Point 1:

I recommend that authors need to read their manuscript carefully before submission. There are so many errors, hence actually I cannot evaluate the manuscript.

Response 1: We are terribly sorry for all the incorrect writings in our previous manuscript. During the revision, we paid more attention on the expression of our manuscript.

Point 2:

Line 12-13 at page 1.

Fenton process, and .?

Response 2: For Reviewer’s kindly comment “Line 12-13 at page 1. Fenton process, and.”, we already rewrite to “PMS/CoMgAl-LDH system overcomes the drawbacks of traditional Fenton processes.” In Line 12-13.

Point 3:

Line 136 at page 5 is not relevant to the content of Table 1.

Response 3: Considering the Reviewer’s suggestion that “Line 136 at page 5 is not relevant to the content of Table 1.”, we realized that it’s better to express directly in the Results and Discussion, like “The BET specific surface area of CoMgAl-LDH catalyst was 92.51 m2/g.” in Line 104.

Point 4:

The condition of the catalytic experiment is improperly written.

Response 4: As Reviewer suggested that “The condition of the catalytic experiment is improperly written.”, we have rewritten the condition parts, like section 3.4.1 “The influence of different catalyst dosage (0, 0.2, 0.3 & 0.4 g/L) for phenol degradation was evaluated in Figure 4. The amount of PMS (3 mM) and phenol (0.1 mM) was fixed.” in Line 163-164.

We made lots of modifications in our manuscript. Special thanks to your sincerely comments.

Reviewer 2 Report

The manuscript by Z. Liao et al. is devoted to the exploration of the catalysts active in ROS (reactive oxygen species) or RSS (reactive sulfur species)-mediated oxidation of the water soluble organic contaminates. As a model contaminant phenol was chosen. The source of reactive species is peroxymonosulfate, and the examined catalysts are based on the layered double hydrotalcites doped with cobalt ions. The topic is very interesting and up-to-date for environmental and water scientist community. The examined catalysts showed excellent activity with the expected stability. A number of factors that could influence catalytic activity was cautiously checked. I find the manuscript interesting, saturated with noticeable data, well interpreted. Yet, some amendments and clarifications are necessary before publication. Also, careful editing, check spelling, and grammatical corrections are necessary. My suggestions for improving the manuscript and critical comments are provided below.

1. Please provide full names for the acronyms already in the abstract.

2. BET is not a characterization technique but a way of analysis of physisorption of N2. Therefore, in paragraph 2.4 low-temperature physisorption of N2 should be mentioned.

3. It is said that the catalysts were characterized with XPS technique (paragraph 2.4), however no results are discussed. It would be important to have some insight into the nature of the active cobalt sites. What is the valence state of the cobalt? Can it be related somehow to the mechanism of PMS activation? What kind of mechanism is assumed?

4. In Table 1 two catalysts are mentioned (CoAl, MgAl) which are not explained in the experimental part. Please include the proper description.

5. What is the possible explanation of the effect of chloride anions on the observed reaction rate? Apparently both catalysts LHD-H and LHD-P behave divergent (Fig. 5C). Maybe some sorption experiments of Cl anions could be of help.

6. Figures 6 and 7 must be redesigned. I suggest preparing two-column figures. It will consume less space and will be more concise.

7. Manuscript must undergo a serious and critical editing. Some sentences cannot be understood. Below there are few examples:

Abstract: “The degradation performance of Co-LDH synthesized by hydrothermal method and co-precipitation method alone with their follow calcinated MMO was evaluated by metal ion leaching and chloride ion effect.”

Page 2. “We recently performed these attractive properties of Co-LDH based catalysts appear outstanding efficiency and stability in bicarbonate activated H2O2 system that associated with diverse free radicals in which the aqueous medium was basic, in contrast to acidic conditions in other traditional Fe2+-Fenton(Jawad et al. 2015).”

Page 2. “Co-LDH catalyst was further studied for the physicochemical properties were characterized…”

and many others to name only few: line 140, lines 253-254.

Author Response

Response to Reviewer 2 Comments

Dear Reviewer:

Thank you for your comments concerning our manuscript entitled “CoMgAl-LDH Synthesized by Hydrothermal Method Catalyzed Peroxymonosulfate with High Stability and Reusability for Degradation of Phenol” (ID: 435643). Those comments are all valuable and very helpful for revising and improving our manuscript, as well as the important guiding significance to our researches. We have studied comments carefully and have made correction which we hope to meet with approval. The main correction in the paper and responds to reviewer’s comments are as flowing:

Point 1:

Please provide full names for the acronyms already in the abstract.

Response 1: As Reviewer suggested that provide full names for the acronyms abstract, we have added full name peroxymonosulfate for PMS, and layered double hydrotalcite for LDH. Like “In this study, we reported an active and stable CoMgAl layered double hydrotalcite (LDH) catalyst for the degradation of phenol by heterogeneous activation of peroxymonosulfate (PMS).” in Line 10-11.

Point 2:

BET is not a characterization technique but a way of analysis of physisorption of N2. Therefore, in paragraph 2.4 low-temperature physisorption of N2 should be mentioned.

Response 2: For Reviewer’s kindly comment, we already rewrite paragraph 2.4, “Surface area of the catalyst was carried out by low-temperature physisorption of N2 on a Micromeritics ASAP 2020 analyzer.” In Line 94-95.

Point 3:

It is said that the catalysts were characterized with XPS technique (paragraph 2.4), however no results are discussed. It would be important to have some insight into the nature of the active cobalt sites. What is the valence state of the cobalt? Can it be related somehow to the mechanism of PMS activation? What kind of mechanism is assumed?

Response 3: As Reviewer suggested that we added the XPS result and discussion in paragraph 3.1. “The oxidation states of the transition metal was related to catalytic activity, XPS was conducted for the oxidation states of Co  (Figure 1 C). The 2P3/2 and 2P1/2 levels appeared at 780.87 eV and 796.49 eV, respectively. The satellite peak at 785.62 eV indicating the Co2+ and Co3+ states existed in CoMgAl-LDH” in Line 104-107.

In paragraph 3.3, we demonstrated sulfate radical was the only oxidant species. XPS analysis conducted for CoMgAl-LDH revealed Co2+ and Co3+ states. In PMS/CoMgAl-LDH system, the activation of PMS via catalyst is proposed as the following steps (Eqs. (1) and (2)).

Co2+ + HSO5- Co3+ + SO4•- + OH- (1)

Co3+ + HSO5- Co2+ + SO5•- + H+ (2)

Point 4:

In Table 1 two catalysts are mentioned (CoAl, MgAl) which are not explained in the experimental part. Please include the proper description.

Response 4: We are very sorry for our negligence of description of CoAl and MgAl. For comparison, CoAl and MgAl catalysts were also prepared by the same procedure as CoMgAl-LDH. In paragraph 3.2 we gave the results of CoAl and MgAl reaction with PMS for phenol degradation, which was much lower than CoMgAl-LDH catalyst. This result clearly demonstrated the excellent phenol removal performance of PMS/CoMgAl-LDH system was attributed to CoMgAl-LDH catalyst.

Point 5:

What is the possible explanation of the effect of chloride anions on the observed reaction rate? Apparently both catalysts LHD-H and LHD-P behave divergent (Fig. 5C). Maybe some sorption experiments of Cl anions could be of help.

Response 5: As other Reviewer’s suggestion, we thought this manuscript should focus on the experimental parameters of PMS/CoMgAl-LDH system in the phenol degradation reaction. So, during the revision, we decide to delete all the data and description of LHD-P and MMO. In our future research, we will study on the effect of preparation method of the catalyst.

Point 6:

Figures 6 and 7 must be redesigned. I suggest preparing two-column figures. It will consume less space and will be more concise.

Response 6: We have made correction according to the Reviewer’s comment. Figure 6 was repositioned next each paragraph of description in Line 172, 182 and 196. In Figure 7, Co2+ leaching at different initial pH condition (C), and pHpzc measurement (D) was deleted, and directly mention in the description. “In acidic condition (pH 3 and 5), the leaching of Co2+ after reaction was 0.39 and 0.13 mg/L, respectively, and in natural or alkaline condition, nearly none Co2+ ( < 0.01 mg/L) could be found in the solution.” in Line 219-221. “The pHpzc of CoMgAl-LDH was valued about 7.7” in Line 210.

Point 7:

Manuscript must undergo a serious and critical editing. Some sentences cannot be understood. Below there are few examples.

Abstract: “The degradation performance of Co-LDH synthesized by hydrothermal method and co-precipitation method alone with their follow calcinated MMO was evaluated by metal ion leaching and chloride ion effect.”

Response 7: We are terribly sorry for all the incorrect writings in our previous manuscript. During the revision, we paid more attention on the expression of our manuscript.

Abstract: “PMS/CoMgAl-LDH system overcomes the drawbacks of traditional Fenton processes. Various effects, e.g., scavengers, chloride ion, catalyst dosage, PMS concentration, temperature and pH were also inspected to evaluate the system.” in Line 12-15.

Point 8:
Page 2. “We recently performed these attractive properties of Co-LDH based catalysts appear outstanding efficiency and stability in bicarbonate activated H2O2 system that associated with diverse free radicals in which the aqueous medium was basic, in contrast to acidic conditions in other traditional Fe2+-Fenton(Jawad et al. 2015).”

Response 8: In Page 2 Line 50-53, we changed this sentence into “Co-LDH catalysts appeared outstanding efficiency and stability in bicarbonate activated H2O2 system, in contrast to acidic conditions of traditional Fe2+-Fenton, H2O2/Co-LDH system demonstrated consistent performance over a broad pH range of 5-12, associated with diverse free radicals.”

Point 9:

Page 2. “Co-LDH catalyst was further studied for the physicochemical properties were characterized…”

and many others to name only few: line 140, lines 253-254.

Response 9: In Page 2 Line 59-60, we rewrite this sentence into “In this work, CoMgAl-LDH catalyst was synthesized by hydrothermal method, and physicochemical properties of the catalyst were characterized.”

Point 10:

and many others to name only few: line 140

Response 10: Previous Line 140 was rewrite into “The phenol degradation performance of PMS/CoMgAl-LDH system was shown in Figure 2.” in Line 114

Point 11:

and many others to name only few: lines 253-254.

Response 11: Previous Line 253-254, all paragraph was rewrite into “As shown in Figure 7 A, the phenol degradation was accelerated at pH 3, 5 and 11 comparing with natural condition (pH=6), but the performance of PMS/CoMgAl-LDH system decreased at pH 9. pKa of HSO5- was 9.4, it can be assumed that HSO5- was the only PMS species in solution at natural and acidic condition (pH 3, 5 and 6), while a small amount of SO52- in the solution at alkaline condition (pH 9 and 11). The pHpzc of CoMgAl-LDH was valued about 7.7. When pH of solution higher than pHpzc of catalyst, the surface of the catalyst was negatively charged. So, the interaction between the catalyst and SO52- decreased the degradation efficiency, when the initial pH was 9. pka of phenol was 10. When the initial pH was 11, phenol existed in its deprotonated form with positive charge, which mean the charge between the catalyst and phenol was mutually exclusive. In theory, the reactivity should be relatively weakened at pH 11, but the performance of PMS/CoMgAl-LDH system enhanced, which indicated that there are other factors to activate PMS beyond CoMgAl-LDH catalyst. We speculated that PMS was activated by alkaline conditions, and this theory was confirmed in Figure 7 B, when initial pH was 11, phenol degradation achieved 100% in 80 min with PMS alone. In acidic condition (pH 3 and 5), the leaching of Co2+ after reaction was 0.39 and 0.13 mg/L, respectively, and in natural or alkaline condition, nearly none Co2+ ( < 0.01 mg/L) could be found in the solution. When the initial pH was 3, only using PMS for phenol degradation was less than 10% (Figure 7 B), the increasing of phenol degradation may due to the presence of PMS/Co2+ system.” in Line 206-223.

We made lots of modification in our manuscript. Special thanks to your sincerely comments.

Reviewer 3 Report

In the manuscript is presented “the degradation of Organic Pollutants and Landfill Leachates” by using cobalt-containing hydrotalcites that were synthesized from two synthetic approaches: hydrothermal and precipitation.

First of all, I have to say that English should be seriously improved because some parts of the manuscript are understandable. Second, it is only studied the degradation of phenol, no the degradation of several pollutants or Landfill leachates and therefore the Title should be changed. Finally, the title refers to the use of samples prepared from the hydrothermal method, but the first results are the comparison between hydrothermal and precipitation based samples. It is not clear which is the main objectives of the paper. Even, the characterization results (very poor) compared the samples prepared from both synthetic approaches and the activity described in section 3.2. However, no correlation structure-activity is performed just to explain the differences observed. Authors should focus the manuscript to the effect of the preparation method of the samples or to the experimental parameters in the degradation reaction.

Moreover, the results are not compared with literature data in order to highlight the research included in the paper and what is really new.

Other things:

·         There are several acronyms in the manuscript not explained

·         Section 2.4. A detailed description of the experimental techniques employed is required

·         The characterization of samples prepared is minimal, even, the comparison of two families of compounds makes no sense if the paper is devoted to experimental factors influencing the photodegradation of phenol

·         XRD: Why do you say that LDH-H presented better crystal structure? Did you measure crystal particle size? There are some non-identified peaks in Figure 1. Figure 1-A clear Identification of all XRD peaks is required.

·         Section 3.2. should include all the photodegradation tests. That section should be reorganized.

·         Phenol degradation-what does MMO-C means in figure 3b?

Author Response

Response to Reviewer 3 Comments

Dear Reviewer:

Thank you for your comments concerning our manuscript entitled “CoMgAl-LDH Synthesized by Hydrothermal Method Catalyzed Peroxymonosulfate with High Stability and Reusability for Degradation of Phenol” (ID: 435643). Those comments are all valuable and very helpful for revising and improving our manuscript, as well as the important guiding significance to our researches. We have studied comments carefully and have made correction which we hope to meet with approval. The main correction in the paper and responds to reviewer’s comments are as flowing:

Point 1:

First of all, I have to say that English should be seriously improved because some parts of the manuscript are understandable.

Response 1: We are terribly sorry for all the incorrect writings in our previous manuscript. During the revision, we paid more attention on the expression of our manuscript.

Point 2:

Second, it is only studied the degradation of phenol, no the degradation of several pollutants or Landfill leachates and therefore the Title should be changed.

Response 2: We are very sorry for our negligence of the title. We already changed the title into “CoMgAl-LDH Synthesized by Hydrothermal Method Catalyzed Peroxymonosulfate with High Stability and Reusability for Degradation of Phenol”

Point 3:

Finally, the title refers to the use of samples prepared from the hydrothermal method, but the first results are the comparison between hydrothermal and precipitation based samples. It is not clear which is the main objectives of the paper. Even, the characterization results (very poor) compared the samples prepared from both synthetic approaches and the activity described in section 3.2. However, no correlation structure-activity is performed just to explain the differences observed. Authors should focus the manuscript to the effect of the preparation method of the samples or to the experimental parameters in the degradation reaction. Moreover, the results are not compared with literature data in order to highlight the research included in the paper and what is really new.

Response 3: It is true as Reviewer suggested that we should focus on only one objective in our paper. So, we decided to focus on the experimental parameters of PMS/CoMgAl-LDH system in the phenol degradation in this manuscript. During the revision, we deleted all the data and description of precipitation-based catalyst. In our future research, we will study on the effect of preparation method of the catalyst. We compared PMS/CoMgAl-LDH system with PMS/Co2+ system reported in literature. “It was reported 0.3 mM Co2+ can strongly active PMS (0.6mM) to generate sulfate radical for degradation 90% of organic pollutants (0.3mM) in water” in Line 116-117. The result indicating that both homogeneous and heterogeneous reactions of Co could efficiently active PMS, but heterogeneous PMS/CoMgAL-LDH system could effectively avoid cobalt ion leaching.

Point 4:

There are several acronyms in the manuscript not explained

Response 4: As Reviewer suggested that provide full names for the acronyms in the manuscript, we have added full name peroxymonosulfate for PMS, and layered double hydrotalcite for LDH. Like “In this study, we reported an active and stable CoMgAl layered double hydrotalcite (LDH) catalyst for the degradation of phenol by heterogeneous activation of peroxymonosulfate (PMS).” in Line 10-11.

Point 5:

Section 2.4. A detailed description of the experimental techniques employed is required.

Response 5: As other Reviewer’s suggestion, Section 2.4 was rewrite in to “The structure of the catalyst was measured by X-ray diffraction (XRD, X’Pert PRO). The morphology of the catalyst was observed from picture taken on a Quanta 200 scanning electronic microscopy. Surface area of the catalyst was carried out by low-temperature physisorption of N2 on a Micromeritics ASAP 2020 analyzer. Metal oxidation states of catalyst was measured with X-ray photoelectron spectroscopy (XPS, V.G scientific ESCALAB mark II system).” in Line 92-96.

Point 6:

The characterization of samples prepared is minimal, even, the comparison of two families of compounds makes no sense if the paper is devoted to experimental factors influencing the photodegradation of phenol.

Section 3.2. should include all the photodegradation tests. That section should be reorganized.

Response 6: As Reviewer’s suggestion, we did more characterization. we used X-ray photoelectron spectroscopy (XPS) to characterize the metal oxidation states of catalyst, the XPS result and discussion in section 3.1. “The oxidation states of the transition metal was related to catalytic activity, XPS was conducted for the oxidation states of Co  (Figure 1 C). The 2P3/2 and 2P1/2 levels appeared at 780.87 eV and 796.49 eV, respectively. The satellite peak at 785.62 eV indicating the Co2+ and Co3+ states existed in CoMgAl-LDH” in Line 104-107.

In section 3.3, we demonstrated sulfate radical was the only oxidant species. XPS analysis conducted for CoMgAl-LDH revealed Co2+ and Co3+ states. In PMS/CoMgAl-LDH system, the activation of PMS via catalyst is proposed as the following steps (Eqs. (1) and (2)).

Co2+ + HSO5- → Co3+ + SO4•- + OH- (1)

Co3+ + HSO5- → Co2+ + SO5•- + H+ (2)

We have reorganized section 3.2, firstly, we compared PMS/CoMgAl-LDH system with PMS/Co2+ system, the result indicating that both homogeneous and heterogeneous reactions of Co could efficiently active PMS, but heterogeneous PMS/CoMgAL-LDH system effectively avoid cobalt ion leaching. Then, we gave the results of Co3O4, MgO, Al2O3, CoAl and MgAl reaction with PMS for phenol degradation, which was much lower than CoMgAl-LDH catalyst. This result clearly demonstrated the excellent phenol removal performance of PMS/CoMgAl-LDH system was attributed to CoMgAl-LDH catalyst.

Point 7:

XRD: Why do you say that LDH-H presented better crystal structure? Did you measure crystal particle size? There are some non-identified peaks in Figure 1. Figure 1-A clear Identification of all XRD peaks is required.

Phenol degradation-what does MMO-C means in figure 3b?.

Response 7: We are very sorry for our negligence of XRD identification. We identified all XRD peaks in Figure 1 A. “Rhombohedral symmetry with well-formed layered structures were demonstrated by (001), (003) and (006) planes at 2θ ≈ 11°, 23°, 35° (JCPDS-022-0700). Peaks at 2θ ≈ 39°, 47°,61° and 62° (JCPDS-089-0460) indicated (015), (018), (110) and (113) crystal planes of LDH compounds respectively, which were similar as reported study”. in Line 99-102.

We are terribly sorry for all the incorrect writings, and MMO was deleted.

We made lots of modification in our manuscript. Special thanks to your sincerely comments.

Round 2

Reviewer 3 Report

Authors have modified the manuscript according to the suggestions made. However, I find some minor points to be considered before publication. 

The title is not clear, it is grammatically incorrect. I really do not understand the meaning.

2.- The description of experimental techniques is minimal. XPS analysis details, deconvolution software should be included.

3.- XPS figure does not include the entire Co 2p spectra. The component Co2p1/2 related to the satellite is not represented. The binding energy of the first Co 2p3/2 peak is due to which Co species? The satellite is related to the presence of Co 2+ ions.

Author Response

Response to Reviewer’s Comments

Dear Reviewer:

Thank you for your comments concerning our manuscript entitled “Degradation of Phenol Using Peroxymonosulfate Activated by a High Efficiency and Stability CoMgAl-LDH Catalyst” (ID: 435643). Those comments are all valuable and very helpful for revising and improving our manuscript, as well as the important guiding significance to our researches. We have studied comments carefully and have made correction which we hope to meet with approval. The main correction in the paper and responds to reviewer’s comments are as flowing:

Point 1:

The title is not clear, it is grammatically incorrect. I really do not understand the meaning.

Response 1: For Reviewer’s kindly suggestion, we already rewrite the title as “Degradation of Phenol Using Peroxymonosulfate Activated by a High Efficiency and Stability CoMgAl-LDH Catalyst”.

Point 2:

The description of experimental techniques is minimal. XPS analysis details, deconvolution software should be included.

Response 2: We are very sorry for our negligence of experimental techniques descriptions, we already rewrite paragraph 2.4, “The structure of the catalyst was measured by X-ray diffraction (XRD, X’Pert PRO) with a diffractometer of Cu Kα radiation (λ= 1.54 Å). The morphology of the catalyst was observed from picture taken on a Quanta 200 scanning electronic microscopy. Surface area of the catalyst was carried out on a Micromeritics ASAP 2020 analyzer, before low-temperature physisorption of N2, catalyst was dried and degassed at 120 °C under vacuum for 1 day. Metal oxidation states of catalyst was measured with X-ray photoelectron spectroscopy (XPS, V.G scientific ESCALAB mark II system) with standard and monochrochromatic source (Al Kα), and binding energiy was calibrated by the carbon signal at 285 eV. XPS spectra was analyzed by XPSPeak 4.1.” In Line 92-99.

Point 3:

XPS figure does not include the entire Co 2p spectra. The component Co2p1/2 related to the satellite is not represented. The binding energy of the first Co 2p3/2 peak is due to which Co species? The satellite is related to the presence of Co 2+ ions.

Response 3:

As Reviewer suggested that we firstly replaced Figure 1 C with the entire Co 2p spectra, then we rewrite the discussion of XPS in paragraph 3.1 as “The oxidation states of the transition metal were related to catalytic activity. XPS was conducted for the oxidation states of Co (Figure 1 C). Co 2P3/2 peak was located at 780.9 eV and with a satellite peak at 785.6 eV. the Spin-orbital splitting between the Co 2p3/2 and Co 2p1/2 peaks is 16.0 eV. According to the Co 2P3/2 binding energy and separation between components of Co 2p doublet, the cobalt-containing material maybe Co(OH)2 [25]. The result indicated the oxidation states of cobalt as Co2+ and excluded the existence of Co3+.” in Line 107-113.
